# Multi-Scale Analysis of the Composition, Structure, and Function of Decellularized Extracellular Matrix for Human Skin and Wound Healing Models

**DOI:** 10.3390/biom12060837

**Published:** 2022-06-16

**Authors:** Atiya M. Sarmin, Nadia El Moussaid, Ratima Suntornnond, Eleanor J. Tyler, Yang-Hee Kim, Stefania Di Cio, William V. Megone, Oliver Pearce, Julien E. Gautrot, Jonathan Dawson, John T. Connelly

**Affiliations:** 1Centre for Cell Biology and Cutaneous Research, Blizard Institute, Queen Mary University of London, London E1 4NS, UK; a.m.sarmin@qmul.ac.uk (A.M.S.); nadiaelmoussaid@hotmail.fr (N.E.M.); ratty.get@gmail.com (R.S.); 2Barts Cancer Institute, Queen Mary University of London, London E1 4NS, UK; e.j.tyler@qmul.ac.uk (E.J.T.); o.pearce@qmul.ac.uk (O.P.); 3Institute of Developmental Sciences, Faculty of Medicine, University of Southampton, Southampton SO17 1BJ, UK; yanghee.kim@soton.ac.uk (Y.-H.K.); jid@soton.ac.uk (J.D.); 4School of Engineering and Materials Sciences, Queen Mary University of London, London E1 4NS, UK; s.dicio@qmul.ac.uk (S.D.C.); w.v.megone@qmul.ac.uk (W.V.M.); j.gautrot@qmul.ac.uk (J.E.G.)

**Keywords:** biomaterials, extracellular matrix, proteomics, skin, biofabrication

## Abstract

The extracellular matrix (ECM) is a complex mixture of structural proteins, proteoglycans, and signaling molecules that are essential for tissue integrity and homeostasis. While a number of recent studies have explored the use of decellularized ECM (dECM) as a biomaterial for tissue engineering, the complete composition, structure, and mechanics of these materials remain incompletely understood. In this study, we performed an in-depth characterization of skin-derived dECM biomaterials for human skin equivalent (HSE) models. The dECM materials were purified from porcine skin, and through mass spectrometry profiling, we quantified the presence of major ECM molecules, including types I, III, and VI collagen, fibrillin, and lumican. Rheological analysis demonstrated the sol-gel and shear-thinning properties of dECM materials, indicating their physical suitability as a tissue scaffold, while electron microscopy revealed a complex, hierarchical structure of nanofibers in dECM hydrogels. The dECM materials were compatible with advanced biofabrication techniques, including 3D printing within a gelatin microparticle support bath, printing with a sacrificial material, or blending with other ECM molecules to achieve more complex compositions and structures. As a proof of concept, we also demonstrate how dECM materials can be fabricated into a 3D skin wound healing model using 3D printing. Skin-derived dECM therefore represents a complex and versatile biomaterial with advantageous properties for the fabrication of next-generation HSEs.

## 1. Introduction

Human skin is a complex organ that performs a wide range of functions, but primarily acts as a barrier to the external environment [1]. It is composed of three distinct layers: the adipose-containing hypodermis, the fibroblast-populated dermis that is also rich with extracellular matrix (ECM), and the stratified squamous epidermis consisting primarily of keratinocytes. Other important features of skin include blood vessels, hair follicles, sweat glands, muscle, and nerves.

Tissue-engineered human skin equivalents (HSEs) have been developed to recapitulate key structural and functional features of native skin [2,3,4]. They are a common alternative to traditional 2D cell culture and animal models and typically consist of an ECM-mimetic dermal scaffold, laden with fibroblasts and covered with a stratified layer of keratinocytes. While these models replicate the basic bi-layered structure of the skin, the barrier function is still less well-developed than native tissue [5,6] and there is a lack of more complex hair follicles and glandular structures. Recent efforts to produce in vitro HSEs that more closely resemble human skin aim to incorporate additional features, such as a hypodermis [7], vasculature [8], or immune cells [9], and make use of advanced biofabrication methods, such as 3D bioprinting [10,11,12].

As the ECM provides essential physical and biochemical signals for regulating skin structure and function, the identification of appropriate biomaterials for mimicking the dermal matrix is a key consideration in HSE development. While previous studies have investigated a wide range of natural polymer hydrogels, such as fibrin [13], type I collagen [11,14], Matrigel [15,16], and gelatin [17], more recent research by our group and others has begun to explore the use of decellularized ECM (dECM) derived directly from the skin [17,18,19]. Decellularization is achieved by removing cells and cellular remnants with detergents while minimizing loss and damage of the native ECM and retaining its composition and ultrastructure as much as possible [20]. Decellularized tissues can also be solubilized by protease treatment and fabricated into stable hydrogels [21]. It is believed that dECM hydrogels are advantageous as they retain the biochemical complexity, biological activity, and nanostructure of the endogenous tissue, but a complete understanding of the composition–function and structure–function relationships of dECM biomaterials remains incompletely understood. While dECM materials are able to form solid gels, their application for 3D bioprinting is also still limited by an overall low viscosity and slow gelation times [22,23].

To improve the utility of dECM biomaterials for skin tissue engineering applications, this study provides an in-depth characterization of the composition, structure, and biomechanics of skin-derived dECM biomaterials. In addition, we extend the potential range of 3D bioprinting applications for dECM-based materials by demonstrating compatibility with several advanced biofabrication techniques and construction of a novel 3D wound healing model.

## 2. Materials and Methods

### 2.1. Human Skin

Redundant human skin was obtained from healthy donors following plastic surgery procedures at the Royal London Hospital or Phoenix Hospital, Essex. All donors provided written informed consent according to local ethical approval (East London Research Ethics Committee, study number 2011-000626-29 and East of England Research Ethics Committee, study number 21/EE/0057). Adipose tissue was removed under aseptic conditions and the skin was cut into 2 cm^2^ pieces.

### 2.2. Preparation of Skin-Derived dECM

Fresh porcine abdominal skin was obtained from a local butcher (London, UK). Under sterile conditions, the skin was prepared by trimming the fat tissue, plucking the hair, and cutting the pieces into roughly 1 cm × 1 cm pieces. The skin was soaked then rinsed three times in 1X PBS with 1% penicillin/streptomycin before being frozen at −80 °C for 5 h and freeze-dried overnight. The tissue was digested overnight at 4 °C in a Dispase II solution (560 U L^−1^) for 2 mL gram^−1^ of dry tissue to disrupt the epidermal–dermal association. Dispase II solution was prepared by diluting in water a stock solution of 50 U mL^−1^ Dispase II (Merck, Darmstadt, Germany, D4693-1G) dissolved in a buffer containing 10 mM sodium acetate (pH 7.5) and 5 mM calcium acetate. The epidermis was then mechanically detached using tweezers, and the dermis was rinsed three times in sterile ddH2O before washing in 70% ethanol overnight at room temperature with constant stirring. The dermis was incubated in 0.25% Trypsin/0.1% EDTA for 1 h at 37 °C. The dermis was washed three times in ddH2O before incubation in a 1% Triton-X 100, 0.26% EDTA, 0.69% Tris buffer solution at room temperature with constant stirring for 6 h and then overnight with fresh Tris buffer solution. The tissue was again washed three times in ddH_2_O, followed by freezing at −80 °C before being freeze-dried overnight. The dry tissue was weighed and digested at 20 mg mL^−1^ in an acidic pepsin solution. The pepsin solution was prepared by dissolving porcine Pepsin A (Merck, Darmstadt, Germany, P7000, 250 U mg^−1^) at a final concentration of 1 mg mL^−1^ in 0.01 M hydrochloric acid (HCl). The porcine dermis was digested at room temperature for 3–5 days under constant stirring until all tissue was digested. The resulting decellularized ECM was aliquoted and stored at −20 °C. Three batches of dECM were produced from three different animals.

The dECM was gelled by mixing one tenth of the digest volume with 1 N NaOH and one ninth of the digest volume with 10X PBS. The mixture was incubated at 37 °C for 1 h to form a soft hydrogel. As a comparison, collagen/Matrigel gels were prepared by making a master mix of 49% Rat Tail Collagen I (3–4 mg mL^−1^) (Corning, Corning, NY, USA, CF715), 21% ice-cold Matrigel Basement Membrane Matrix, LDEV-Free (354234, Corning), 10% 10X Minimum Essential Medium (MEM) (11430030, Thermo Fisher Scientific Life Technologies, Horsham, UK), and 10% fetal bovine serum (FBS). Each component was added in and slowly mixed by pipetting. The pH was adjusted to 7.4 by adding 1 N NaOH in a drop-wise fashion.

All reagents were from Sigma-Aldrich unless otherwise stated.

### 2.3. Mass Spectrometry Sample Preparation

Three batches of dECM samples from three different animals were prepared for mass spectrometry as described previously by Naba et al. [24]. Triplicate samples of three independent batches of dECM hydrogel were solubilized in 8 M urea and 20 mM HEPES containing Na_3_VO_4_ (100 mM), NaF (0.5 M), β-glycerol phosphate (1 M), and Na_2_H_2_P_2_O_7_ (0.25 M) at 80 μg of protein for 300 µL of lysis buffer. Samples were vortexed for 30 s and held on ice prior to sonication at 50% intensity, 3 times for 15 s. These resuspended and partially solubilized ECM-enriched samples were directly digested into peptides. 

Disulfide bonds were reduced by adding 5 mM dithiothreitol to the samples for a 1 h incubation period under agitation at room temperature. Free cysteines were alkylated by adding iodoacetamide at a final concentration of 8.3 mM for 1 h under agitation at room temperature in the dark. After diluting to 2 M urea and 20 mM HEPES, samples were deglycosylated with 1500 units of PNGaseF (New England BioLabs, Ipswich, MA, USA) by incubation at 37 °C for 2 h under agitation. Samples were predigested with 1.6 μg of Lys-C (Thermo Fisher, Horsham, UK), incubated at 37 °C for 2 h under agitation, and digested with bead-immobilized trypsin (40 μL of beads per 250 μg of protein; Thermo Fisher) for 16 h at 37 °C under agitation. Samples were acidified with trifloroacetic acid (TFA, 1% *v*/*v*), centrifuged at 2000× *g* for 5 min at 5 °C. Supernatants were transferred to clean microcentrifuge tubes on ice. Glygen TopTips were equilibrated with 100% LC–MS-grade acetonitrile (ACN) and then washed with 99% H_2_O (+1% ACN, 0.1% TFA) prior to loading the peptide samples. Samples were washed with 99% H_2_O (+1% ACN, 0.1% TFA) and desalted peptides were eluted with 70/30 ACN/H_2_O + 0.1% TFA. Desalted peptides were dried and stored at −20 °C.

### 2.4. Liquid Chromatography−Tandem Mass Spectrometry (LC−MS/MS)

Equal volumes of each sample, corresponding to ∼100 ng of material, were initially analyzed via LC−MS/MS, and the total numbers of peptides identified and the sum of the intensities of the precursor ions were used as a normalization metric to determine equivalent peptide amounts. Peptides were separated by reversed-phase HPLC, using an EASY-nLC1000 (Thermo Fisher) with a pre-column (made in house, 6 cm of 10 μm C18) and a self-pack 5 μm tip analytical column (12 cm of 5 μm C18, New Objective, Littleton, MA, USA) over a 140 min gradient, before nanoelectrospray using a QExactive mass spectrometer (Thermo Fisher). Solvent A was 0.1% formic acid, and solvent B was 80% MeCN/0.1% formic acid. The gradient conditions were 0–10% B (0–5 min), 10–30% B (5–105 min), 30–40% B (105–119 min), 40–60% B (119–124 min), 60–100% B (124–126 min), 100% B (126–136 min), 100–0% B (136–138 min), and 0% B (138–140 min). The mass spectrometer was operated in a data-dependent mode. The parameters for the full MS scan were: resolution of 70,000 across 350–2000 *m*/*z*, AGC 3e6, and maximum IT at 50 ms. The full MS scan was followed by MS/MS for the top 10 precursor ions in each cycle with a normal collision energy of 28 and dynamic exclusion of 30 s.

### 2.5. Protein Identification

Raw mass spectral data files (raw) were searched using Proteome Discoverer (Thermo Fisher) and Mascot version 2.4.1 (Matrix Science, London, UK) using the SwissProt Homo sapiens database containing 20,199 entries. Mascot search parameters were: 10 ppm mass tolerance for precursor ions; 15 mmu for fragment-ion mass tolerance; two missed cleavages of trypsin; fixed modification was carbamidomethylation of cysteine; and variable modifications were oxidized methionine, deamidation of asparagine, pyro-glutamic acid modification at N-terminal glutamine, and hydroxylation of lysine and proline. Only peptides with a Mascot score ≥25 and an isolation interference ≤30 were included in the data analysis. Confidently identified proteins were further annotated as being part of the extracellular matrix, as previously defined [25,26], using the tool Matrisome Annotator. In addition, keratins were manually identified and annotated.

### 2.6. Western Blot Analysis

The dECM was mixed with 8 M urea, pH 7.0, and sonicated to reduce viscosity for 15 cycles on (30 s) and 15 cycles off (30 s) at 4 °C. The lysates were mixed with 4X NuPage Loading Buffer (NP0008, Thermo Fisher) mixed with 1% β-mercaptoethanol before being boiled at 90 °C for 10 min. Lysates were separated by 8–12% (*w*/*v*) sodium dodecyl sulphate polyacrylamide gels electrophoresis (SDS-PAGE) (Bio-Rad, Watford, UK) at 100 V for between 1 and 1.5 h in running buffer (0.4 M tris, 0.2 M glycine, 10% SDS dissolved in ddH20 water). The separated proteins were transferred to nitrocellulose membranes by electroblotting the gel with the membrane at 3 Amp for 1 h in transfer buffer (0.4 M tris and 0.2 M glycine dissolved in distilled deionized water). Membranes were blocked in milk buffer (5% (*w*/*v*) non-fat milk in 1X Tris-Buffered Saline, 0.1% Tween (TBS-T)) for 1 h at room temperature. After washing the blocked membranes three times in 1X TBS-T, the proteins were probed with primary antibody, 1:1000 collagen I (mouse, ab90395, Abcam, Cambridge, UK), 1:1000 collagen III (mouse, ab7778, Abcam), or 1:1000 collagen VI (mouse, ab6588, Abcam), diluted in 5% milk TBS-T overnight at 4 °C. The membranes were washed three times in TBS-T, then were incubated in Horse Radish Peroxidase (HRP) conjugated secondary antibody, 1:5000 anti-mouse (P044701-2, Agilent, Santa Clara, CA, USA) and 1:5000 anti-rabbit antibodies (P044801-2, Agilent). The membranes were developed by a 30 s incubation in a chemiluminescence (ECL) substrate (Thermo Fisher, 32106). The protein bands were visualized by a ChemiDoc XRS+ (BioRad).

### 2.7. DNA Quantification

The Quant-iT^™^ PicoGreen^™^ dsDNA Assay Kit (L3224, Thermo Fisher) was used to assess the cellular content of the dECM materials. Standards were prepared using serial dilutions of Calf Thymus DNA (10 mg/mL) in 1X Tris-EDTA buffer (20 mM Tris-HCl; 1 mM EDTA, pH 6.5), 50 ng mL^−1^ of DNA being the highest standard concentration. In a 96 transparent well-plate, a Picogreen solution was added to all the dECM and standard samples and read with a CLARIOstar^®®^ Plus (BMG Labtech, Aylesbury, UK) microplate reader at 420 nm excitation and 520–530 nm emission.

### 2.8. Scanning Electron Microscopy

dECM and collagen I/Matrigel hydrogels were crosslinked at 37 °C for 1 h before being fixed with 2.5% glutaraldehyde in 1X PBS for 2 h at room temperature on a shaker. Human skin tissue was also fixed with 2.5% glutaraldehyde in 1X PBS for 2 h at room temperature on a shaker. Samples were then washed 3 times with 1X PBS and dehydrated with a series of ethanol washes, using ethanol concentrations of 20%, 30%, 40%, 50%, 70%, 90%, and 100%, whereby each wash repeated twice for 5 min. Critical point drying was then performed using an EMS 850 Critical Point Dryer. Samples were characterized via scanning electron microscopy (Inspect F from FEI, Hillsboro, OR, USA).

### 2.9. Rheological Analysis

Rheological measurements were carried out on a Discovery HR-3 Rheometer (TA-Instruments, Wilmslow, UK). Analysis was performed using a steel 40 mm parallel plate top geometry and a steel Peltier plate as the bottom geometry. The gap height was brought to 300–1000 μm and the material was loaded between the top and bottom temperature-controlled rheometer plate. Pluronic F127 hydrogel at a 25% polymer concentration was tested and used as a reference material due to its shear-thinning behavior. Any excess material was trimmed with a spatula before experiments began. Viscosities were determined by flow sweeps performed from shear rates of 1.0 to 5000.0 s^−1^ with 5 points per 10.0 s, a 10.0 s sample period, a 120 s maximum equilibration time, and a 5.0% tolerance. After performing these experiments, the material was replaced with a fresh sample for the following thermosensitive experiments. To measure the gelation of the dECM upon raising the temperature, the storage and loss moduli were probed with oscillatory shear strain of 1.0%, a 10.0 radians second^−1^ angular frequency with a start temperature of 20 °C to an end temperature of 40 °C, and a 2 °C per min temperature step. Finally, samples were characterized by initially applying a peak hold flow sweep at a shear rate of 50 s^−1^ and recording the applied torque. Repeated creep tests were then performed applying the obtained torque for 15 s, then releasing for 10 s and repeating this for 20 cycles.

### 2.10. Extrusion-Based 3D Printing

dECM was mixed with one tenth of the digest volume of 1 N NaOH and one ninth of the digest volume of 10X PBS and warmed to room temperature. The neutralized dECM was loaded into a 3 CC cartridge using a stopper. A 3 CC piston was inserted into the cartridge and the air was pushed out. A G27 needle with a 0.22 mm diameter and 6.35 mm length (DD0135N, NDD-G27L6.35, RegenHU, Villaz Saint-Pierre, Switzerland) was attached to the cartridge. The loaded cartridge was fitted to the extrusion 3D Discovery (RegenHu) printhead and attached to the pneumatic system with a 3 CC adapter. The 3D model was designed on BioCAD (RegenHu) software, converted to G-code, and uploaded to the printer. The pressure was adjusted to approximately 0.02–0.04 MPa, the required amount for continuous extrusion of the dECM, before the program was run and the dECM was extruded at room temperature. For dECM + gelatin materials, the pressure was adjusted to approximately 0.1 MPa and printing was performed at 29 °C using a temperature control jacket (RegenHU) on the extrusion print head. For all experiments, the print bed was maintained at room temperature.

### 2.11. Gelatin Slurry Support Bath Printing

A base solution of 50 mM HEPES and 50 mM NaHCO_3_ dissolved in 300 mL deionized water was prepared. Gelatin (Type A, porcine skin, G8150, Sigma-Aldrich, Darmstadt, Germany) was dissolved in the base solution to a final concentration of 4.5% (*w*/*v*) and the gelatin solution was heated at 50–60 °C until fully dissolved, before allowing the solution to gel overnight at 4 °C. A total of 1.5 vol of the base solution was mixed with the gelatin gel and frozen at −20 °C for 30 min. Next, this mixture was blended in a Venga VG BL 3009 blender (450 Watts) for 120 s until a slurry of small particles was achieved. The slurry was aliquoted into 25 mL aliquots in 50 mL tubes for centrifugation for 2 min at 3000× *g* at 4 °C. The supernatant was discarded, and the remaining slurry was further mixed with 25 mL cold base solution. This mix was vortexed to detach the gelatin particles and was centrifuged three times for a further 2 min, until the supernatant was clear. This solution was stored at 4 °C and used for printing within 2 weeks.

For printing, the neutralized dECM was mixed with CaCl_2_ to a final concentration of 0.1 M to be printed in the gelatin microparticle slurry. The gelatin slurry was transferred into a 60 mm petri dish, a container of sufficient size to hold the part to be printed, and was chilled at 4 °C overnight to remove any trapped air bubbles. The dECM was extruded at room temperature into the support bath using a 27 G needle with a 12.70 mm needle length (927050-TE, Metcal, Chandler’s Ford, UK).

### 2.12. Construction of Wound Healing Model and Sacrificial 3D Printing

A 3D printed frame for the wound healing model was fabricated by first mixing DOWSIL SE1700 silicone (Dow Europe GMBH) at a 10:1 ratio of base to curing agent by weight and removing air bubbles in a vacuum desiccator for 10–15 min. The SE1700 was then transferred to a 3 CC cartridge and extruded at room temperature from a 27G needle at 0.2–0.6 MPa in the pattern designed on BioCAD. The frames were cured at 80 °C overnight and sterilized by autoclaving. Pluronic F127 (25% *w*/*v*) was 3D printed into the center of the frame using a 27G needle and 0.6–1 MPa pressure at room temperature. Pluronic rings of varying dimensions were printed 10 layers high. A cell suspension of dermal fibroblasts was labeled with CellBrite Green according to the manufacturer’s instructions and combined with neutralized dECM and 2 mg mL^−1^ fibrinogen at a density of 4 × 10^5^ cells mL^−1^, then pipetted around the Pluronic ring. The dECM was gelled at 37 °C for one hour, and the Pluronic ring was dissolved and flushed away with cold DMEM. THP-1 monocytes were labeled with CellBrite Red, suspended in a 20 mg mL^−1^ fibrinogen solution, combined 1:1 with 20 U mL^−1^ thrombin, and then pipetted into the wound defect to gel. The constructs were then submerged in DMEM plus 10% FBS and 1% pen/strep for subsequent culture.

For non-wounded HSE constructs, fibroblasts were combined with neutralized dECM alone and gelled in the inner silicone chamber without a Pluronic ring. NTERT keratinocytes were then seeded on the surface of the gel and the culture was submerged for 24 h in FAD medium. Constructs were then raised to the air–liquid interface by removing the medium in the inner chamber and replacing the medium in the outer chamber with 550 μL. HSEs were cultured for 14 days prior to fixation and histological analysis.

### 2.13. Cell Culture

All cell culture reagents were from Thermo Fisher unless otherwise stated. Telomerase immortalized keratinocytes (NTERTs) were cultured at 37 °C with 5% CO_2_ in complete FAD medium containing 1 part Ham’s F12 and 3 parts Dulbecco’s Modified Eagle medium (DMEM) +GlutaMAX, pyruvate. The base media was supplemented with 10% FBS, 1% pen/strep, 10^−10^ M cholera toxin (C8052, Sigma-Aldrich) 5 μg mL^−1^ insulin (Sigma-Aldrich), 10 ng mL^−1^ mouse epidermal growth factor (EGF) (Peprotech, London, UK), 0.5 μg mL^−1^ hydrocortisone, and 1.8 × 10^−4^ M adenine (Sigma-Aldrich). 

Primary dermal fibroblasts (passage 1–4) isolated from redundant skin from plastic surgery procedures were kindly provided by Prof. Michael Philpott and were cultured at 37 °C with 5% CO_2_ in complete DMEM +GlutaMAX medium supplemented with 10% FBS and 1% pen/strep. 

THP-1 monocytes were purchased from ATCC and maintained in RPMI-1640 supplemented with 0.05 mM beta-mercaptoethanol, 10% FBS, and 1% pen/strep. Cells were cultured in suspension at 37 °C with 5% CO_2_.

### 2.14. Histology and Immunofluorescence Staining

The fixed HSE tissue sections were dehydrated and wax-embedded before being sliced into 7 μm sections using a microtome. The sections were dewaxed in xylene, hydrated in 100%, 90%, and 70% ethanol, stained with Harris Haematoxylin (Poly Scientific, Bay Shore, NY, USA) for 5 min, washed with running water for 5 min, rinsed with 1% hydrochloric acid in ethanol for 5 s, washed with running water for 5 min, stained with 0.5% eosin for 5 min, washed with distilled water for 5 min, then dehydrated in 70%, 90%, and 100% ethanol and xylene before being mounted with DPX mounting medium (Sigma-Aldrich) in a fume hood to dry overnight. Images were captured on an Eclipse 80i Stereology Microscope, (Nikon, Tokyo, Japan) using a 10X objective.

For immunofluorescence staining, the fixed tissue (HSE models, human normal skin tissue) sections were dewaxed in xylene and hydrated in 100%, 90%, and 70% ethanol. Antigen retrieval was performed by boiling the tissue sections in a sodium citrate buffer (10 mM sodium citrate, pH 6.0) at 100 °C for 15 min. The slides were cooled and rinsed in water before being permeabilized with 0.2% Triton-X 100 in 1X PBS for 10 min. Tissue sections were incubated in a blocking buffer (1X PBS, 1% BSA (*w*/*v*), 2% FBS (*v*/*v*)) for 1 h at room temperature. Sections were then washed three times with 1X PBS before being incubated overnight in primary antibodies diluted in blocking buffer at the following concentrations: 1:250 keratin 14 (K14), mouse (ab7800, Abcam); 1:250 transglutaminase-1 (TGM-1), rabbit (HPA040171, Atlas Antibodies, Bromma Sweden). Next, the sections were rinsed with 1X PBS and were incubated in secondary antibodies diluted in blocking buffer: 1:1000 Alexa 568 Anti-mouse, donkey (Molecular Probes, Eugene, OR, USA); 1:1000 Alexa 488 Anti-rabbit, donkey (Molecular Probes). To prepare the samples for imaging, mounting medium with T4′,6-diamidino-2-phenylindole (DAPI) (ab104139, Abcam) was dropped on top of each tissue section and 24 × 60 mm coverslips were mounted on top. Mounted samples were left to dry in the dark. Tissue sections were then imaged on a DM4000 epifluorescent microscope (Leica, Wetzlar, Germany) using a 10X objective.

### 2.15. Statistical Analysis

All results are expressed as means ± standard error of the mean (SEM). The differences between two groups were compared by the unpaired *t*-test. *p* values < 0.05 were considered statistically significant. All analyses were performed on Graphpad Prism 9.0.0.

## 3. Results

### 3.1. Analysis of dECM Composition and Structure

Skin-derived dECM biomaterials were prepared according to established protocols within our laboratory [19]. Porcine skin from the abdomen was decellularized using Dispase II and a combined Trypsin/Triton X-100 treatment, followed by digestions with Pepsin A. The dECM digest could then be formed into a soft hydrogel by neutralization with NaOH and incubation at 37 °C. Porcine skin was selected for these studies as it is highly similar to human skin and readily available in large quantities [27,28,29] and our previous studies have demonstrated that these materials support excellent epidermal differentiation and stratification in 3D HSE models [20].

To characterize the complex composition of the dECM materials, we performed proteomic analysis on the solubilized ECM using mass spectrometry (MS) [24]. MS analysis revealed the abundance of type I collagen and type III collagen, the most prominent fibrillar collagens in the dermis (Figure 1a,b). We also observed the presence of the micro-fibrillar type VI collagen, proteoglycans (lumican), elastic fiber proteins (fibrillin), and some residual keratins (Figure 1c and Appendix A). This composition was similar across three independent batches of dECM from different animals, indicating the reproducibility of the decellularization process (Figure 1b,c). These analyses demonstrated that our porcine skin-derived dECM was mainly comprised of dermal collagens and had retained additional matrisome molecules. 

To validate the MS analysis, we performed Western blot analysis of types I, III and VI collagen, and compared the dECM with the standard type I collagen/Matrigel (Col I/Mat) hydrogel used for skin organotypics (Figure 1d) [30]. We observed an abundance of type I collagen that had been retained in the material during the decellularization process, with strong bands at 200 kDa and 120 kDa, representing non-denatured type I collagen and its isoform. We also observed an abundance of type III collagen in both matrices, and type VI collagen only in the dECM. Importantly, we found a similar abundance and comparable banding patterns of these collagens across the three batches of dECM, further demonstrating the reproducibility of the skin decellularization process.

To analyze the efficiency and success of the decellularization, we performed a DNA quantification assay and measured less than 30 ng mL^−1^ of DNA in the 20 mg mL^−1^ dECM solution (Figure 1e), corresponding to approximately 1.5 ng DNA per mg dry mass and >99.9% removal of DNA content [31]. Based on in vivo studies of decellularized material that avoided adverse cell and host responses, less than 50 ng of dsDNA per mg of dry mass is required for successful decellularization [32,33]. Thus, our protocol provides efficient decellularization of porcine skin. 

To assess the nanostructure of crosslinked dECM, we imaged dECM gels using scanning electron microscopy (SEM) and compared against crosslinked Col I/Mat and human skin tissue. SEM images illustrate the retention of ECM architecture and preservation of organized fibrillar bundles in both Col I/Mat and dECM gels (Figure 2). The dECM gel displayed a hierarchical structure of large banded collagen fibers interspersed with fine membranous matrix, which is comparable to the nanostructure of human skin. By contrast, Col I/Mat gel appeared more homogeneous with slightly smaller collagen fibers. The complex structure of the dECM may be due to the presence of additional ECM proteins, such as type VI collagen and fibrillin, making it a better mimic of human skin than Col I/Mat. In addition, proteoglycans, such as lumican (Appendix A), present in the dECM hydrogel may help confer structure to the dermal matrix by organizing collagen and elastic fibers [34].

### 3.2. Analysis of dECM Mechanical Properties

To evaluate the mechanical properties, we performed rheological analyses of the viscoelastic and thermo-responsive properties of the dECM biomaterials and compared them against the Col I/Mat standards, as well as analyzing a shear-thinning material, Pluronic F127, used widely in 3D printing applications. Storage and loss moduli and viscosity were measured at increasing temperatures between 20–40 °C at 2 °C increments. For dECM, the ratio of the loss modulus to the storage modulus (G”/G’), represented by tan(ẟ), was greater than one up to 34 °C and the G” and G’ crossed over at 34 °C (tan(ẟ) = 1) (Figure 3a). This behavior indicates that the dECM hydrogel has a sol-gel transition temperature of approximately 34 °C and is more elastic than viscous at physiologic temperatures. This response was paralleled in the Col I/Mat gel but not for Pluronic F127, as expected (Figure 3b,c). There was also a slight decrease in viscosity of the dECM with increasing temperature (Appendix A). Using flow sweep experiments, which demonstrate the viscosity as a function of shear rate, we observed a shear-thinning behavior of the dECM materials whereby viscosity decreases with increasing shear rates (Figure 3d–f). We also confirmed that the dECM viscosity and shear-thinning behavior was recoverable over repeated applications of shear stress (Appendix A). While shear-thinning is an advantageous property for 3D printing applications, the dECM had a viscosity approximately four orders of magnitude lower than Pluronic F127, suggesting that the viscosity of dECM on its own may be too low for 3D printing with high shape fidelity.

Using the G’ value as an estimate of the elastic shear modulus and assuming a Poisson’s ratio of 0.5, we calculated the Young’s modulus (E) of the dECM to be approximately 6 Pa (E = 2G × (1 + v), where v is Poisson’s ratio). The dECM is therefore substantially softer than the dermis of human skin, which has an elastic modulus of 8 kPa [35], and this mismatch may be an important consideration for future applications in tissue engineering.

### 3.3. Application in Advanced Biofabrication Methods

To demonstrate the flexibility of the dECM biomaterial, we next explored its compatibility with 3D printing and advanced fabrication methods. On its own, dECM displayed poor shape fidelity when printed into a simple ring structure, as expected from its low viscosity (Figure 4a). To improve printability, we first blended the dECM with 20% gelatin at a 1:1 ratio, which increased the viscosity while maintaining a shear-thinning behavior at 30 °C (Appendix A). The dECM-gelatin blend could be 3D printed as a free-form ring structure and held its shape following gelation (Figure 4a). 

Next, we printed a ring-shaped template using Pluronic F127 as a sacrificial material [36,37,38]. The dECM hydrogel was cast within the Pluronic template and gelled at 37 °C (Figure 4b). The Pluronic F127 template was then removed with cold PBS, only leaving behind the ring-shaped dECM structure (Figure 4b).

Finally, we utilized a microparticle slurry as a temporary, thermo-reversible support bath that can be washed away after the printing and gelling of the dECM bioink [39]. The dECM was extruded into a 4.5% gelatin microparticle support bath, which behaved as a Bingham plastic material and maintained the intended structure during the printing process (Figure 4c). During the incubation period where the dECM formed a gel, the gelatin slurry was melted and flushed out with PBS (Figure 4c). Here, we observed the production of intact dECM rings or more complex written structures following removal of the support bath (Figure 4c). Together, these approaches extend the range of potential applications for dECM biomaterials and demonstrate that skin-derived dECM materials are compatible with multiple biofabrication and 3D printing methods.

### 3.4. Fabrication of 3D Wound Healing Models Using Skin dECM

To further demonstrate the applicability of the dECM in advanced 3D printed in vitro models, we constructed a prototype 3D wound healing model using Pluronic F127 as a sacrificial wound template (Figure 5a). To support the model, we first printed an inner and outer silicone (SE1700) frame. The inner frame retained the dECM hydrogels, while the outer frame held the medium and was connected to the dECM via openings in the base of the inner frame. This design allowed for separation of the medium from the surface of the dECM to create an air-liquid interface. We confirmed that this system could be used to generate stratified and well differentiated epidermal layers in skin equivalent models (Appendix A), consistent with our previous results [19].

To create a wound template within this platform, we 3D printed a sacrificial Pluronic ring, and the dECM hydrogel mixed with a low concentration of fibrinogen (2 mg mL^−1^) was cast and gelled around the ring. The Pluronic ring was then removed by flushing with cold (4 °C) medium. A fibrin gel was then cast in the defect to act as a model wound bed, and this construct could be cultured with cell medium in the outer frame prior to downstream analyses. To demonstrate the flexibility of this model, we generated wounds of varying sizes (0.5–3 mm diameter) and shapes (circles and ellipses; Figure 5b). As proof-of-principle, we also seeded THP-1 monocytes in the fibrin wound and dermal fibroblasts in the dECM gels and observed close approximation of the two compartments with precise patterning of the cells (Figure 5c). Thus, the model is compatible with patterning of different cells types in the wound bed and surrounding tissue. Overall, this system represents a flexible and tunable platform for modeling wound healing in 3D and a novel application of dECM-based biomaterials.

## 4. Discussion

In this study, we characterized a skin-derived dECM biomaterial, which displays a complex composition and nanoscale structure and supports the development of HSE and wound healing models. The dECM was extracted from dermal porcine tissue by enzymatic digestion and decellularization, leaving primarily the ECM components while removing most of the cellular content. The removal of DNA from animal-derived biomaterials has been shown to be important for avoiding immune responses [22] while retaining ECM components is necessary to mimic the native tissue environment [18]. Our results confirm that the decellularization of dermal tissue is successful, and there is little batch to batch variation with regards to ECM content. This result is comparable to previous work where a dECM bioink was produced from porcine skin and was shown to retain structural and functional ECM proteins, including collagens, glycosaminoglycans (GAGs), and growth factors [40].

The proteomic analysis performed in this study provides new insight into the complex composition of skin-derived dECM. We demonstrate the presence of essential ECM proteins, such as types I and III collagens. Additionally, the dECM contains the glycoprotein fibrillin, which may be important for modulating and transforming growth factor (TGF)-β and bone morphogenetic protein (BMP) signaling, and for regulating ECM deposition and remodeling [41]. Furthermore, the presence of type VI collagen in the dECM and not in the Col I/Mat may influence the organized structure of the dECM gel due to its role in regulating matrix assembly [42]. The organized assembly of fibers in the dECM hydrogel also indicates that a more complex ECM nanostructure was obtained, and is more comparable to the nanostructure of human skin than Col I/Mat [43].

While the viscosity and shear moduli of the dECM materials are quite low, they are consistent with similar methods for preparation of skin-derived dECM [43], and the work presented here and elsewhere suggests that blending dECM with other biomaterials, such as fibrinogen or gelatin, can improve their biomechanical properties. Future studies should also consider a systematic analysis of the relationship between decellularization protocol parameters, such as digestion times and dECM concentration, and rheological properties in order to standardize these methods.

In this paper, we also present new applications of the dECM biomaterial in a variety of advanced biofabrication methods. For example, Pluronic F127 was used as a sacrificial material and template to create 3D dECM structures, but in future work it may be possible to use Pluronic to engineer microfluidic channels or vascular networks within dECM hydrogels [44,45]. In addition, we demonstrate the compatibility of dECM with 3D printing in a gelatin support bath, which may be advantageous for generating complex 3D structures [39,46]. We also report the printability of the dECM when blended 1:1 with another biomaterial, 20% gelatin, which is significantly higher than the previously reported 1.5–3% dECM composition of bioinks [40,47], and may allow for direct free-form printing of dECM-containing materials. Finally, proof-of-concept studies demonstrate the utility of dECM materials in the construction of 3D wound healing models, and in future studies we aim to use this platform to study the complex cellular interactions, such as fibroblasts and immune cells, that regulate wound healing.

Overall, skin-derived dECM hydrogels display advantageous physical and biological properties for applications in skin tissue engineering. These biomaterials therefore have the potential to be used as dermal matrices for in vitro models of normal, diseased, or wounded human skin, scaffolds for tissue repair, or as bioinks for 3D bioprinting applications. 

## 5. Conclusions

The findings of this study provide an in-depth and more complete characterization of the composition, structure, and mechanical properties of skin-derived dECM biomaterials than was previously available. We demonstrate that dECM materials retain a complex mixture of fibrillar and micro-fibrillar collagens, proteoglycans, and glycoproteins, alongside a hierarchical nanofibrous structure. In addition, dECM biomaterials are compatible with advanced biofabrication methods and 3D wound healing models. Thus, skin-derived dECM is a versatile biomaterial with advantageous properties for skin modeling and tissue regeneration applications.

## Figures and Tables

**Figure 1 biomolecules-12-00837-f001:**
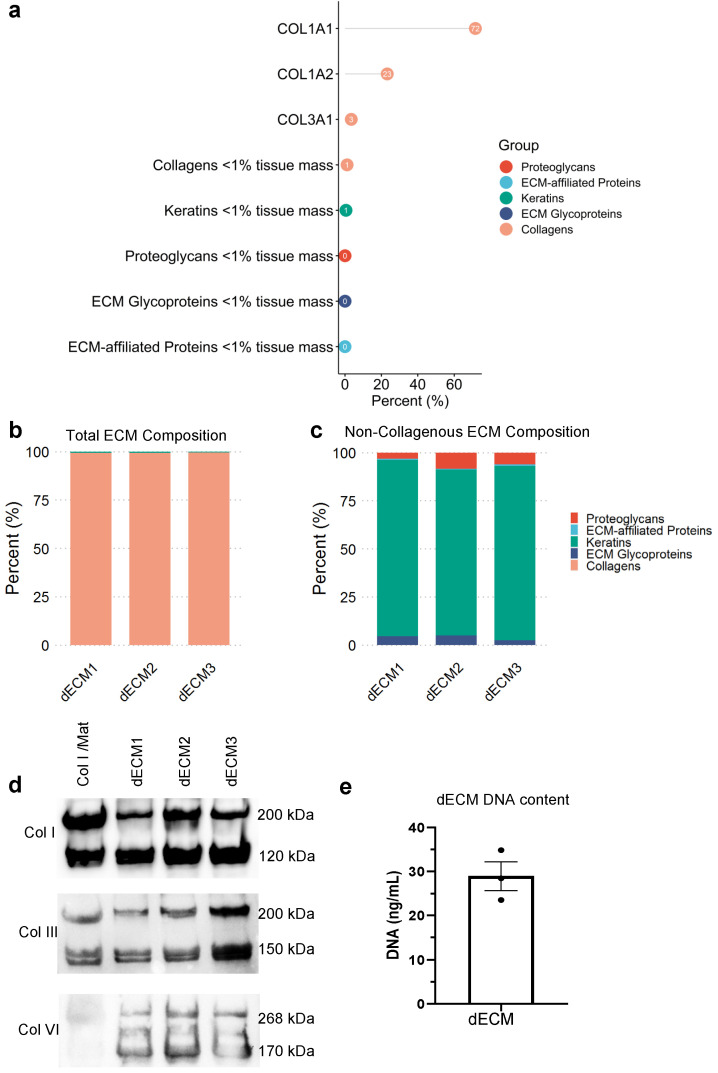
dECM protein composition and DNA content. (**a**) Mass spectrometry analysis of the percent abundance of matrisome proteins in the dECM in the order of abundance as an average of two technical repeats of dECM from three animals. The percent abundance of matrisome proteins across the three batches of dECM, including (**b**) and excluding (**c**) the dominating collagen proteins. (**d**) The collagen content assessed and validated using Western blot analysis, demonstrating the presence of types I, III, and VI collagen in 10 mg mL^−1^ of dECM compared to a collagen I/Matrigel matrix. (**e**) Quantification of residual DNA content using PicoGreen assay. Error bar represents standard error of the mean (SEM) of three technical replicates of one batch of dECM.

**Figure 2 biomolecules-12-00837-f002:**
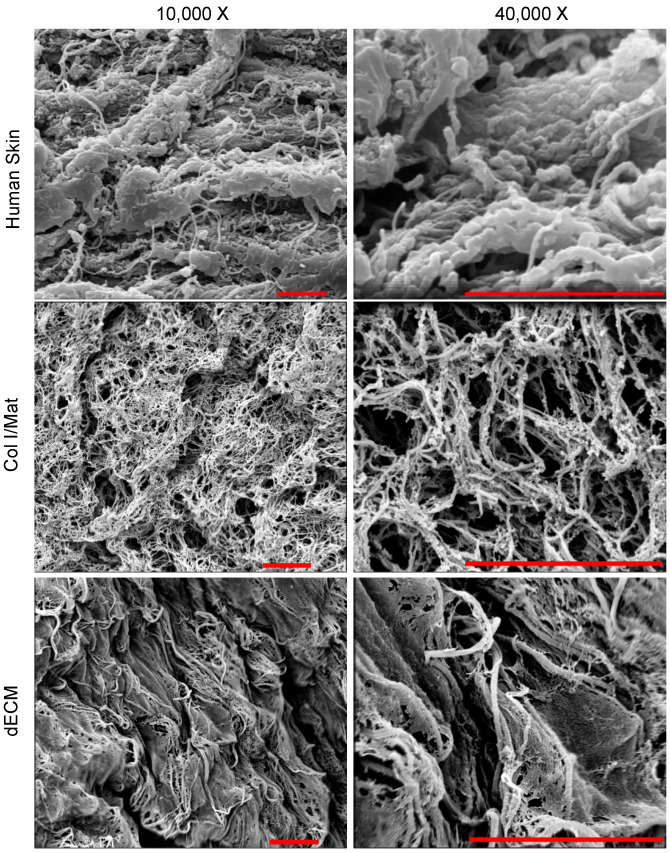
Nanostructure of human skin, and Col/Mat and dECM hydrogels. Human skin images were captured from fresh human abdomen skin. Col/Mat and dECM hydrogels were neutralized and cured at 37 °C for one hour before being fixed and processed for scanning electron microscopy. Scanning electron micrographs were obtained at 10,000× and 40,000× magnification for both gels. Scale bars represent 4 μm.

**Figure 3 biomolecules-12-00837-f003:**
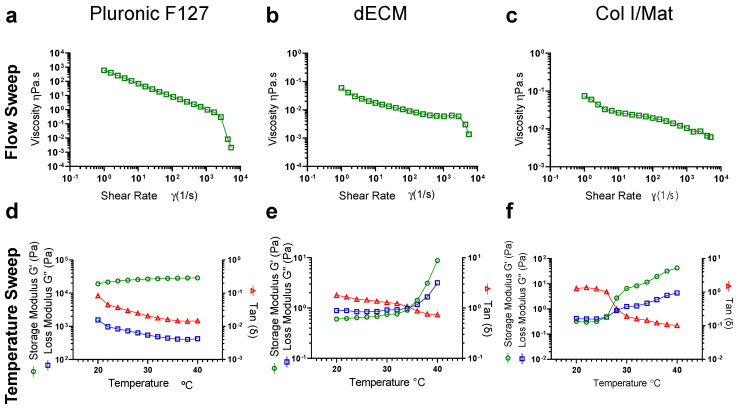
Rheological properties of the dECM and collagen I/Matrigel materials. (**a**–**c**) Flow sweep and (**d**–**f**) temperature sweep experiments were performed on dECM (20 mg mL^−1^) and collagen I/Matrigel (Col I/Mat) versus 25% Pluronic F127. Experiments were carried out on a DHR3 Rheometer (TA Instruments), with dECM materials pre-warmed at 37 °C and neutralized with 1 N NaOH and 10X PBS and with Col I/Mat materials cooled at 4 °C and neutralized with 1 N NaOH. Pluronic F127 data represents the average of four repeat experiments. dECM data represents the average of three independent experiments. Col I/Mat represents the average of two independent experiments.

**Figure 4 biomolecules-12-00837-f004:**
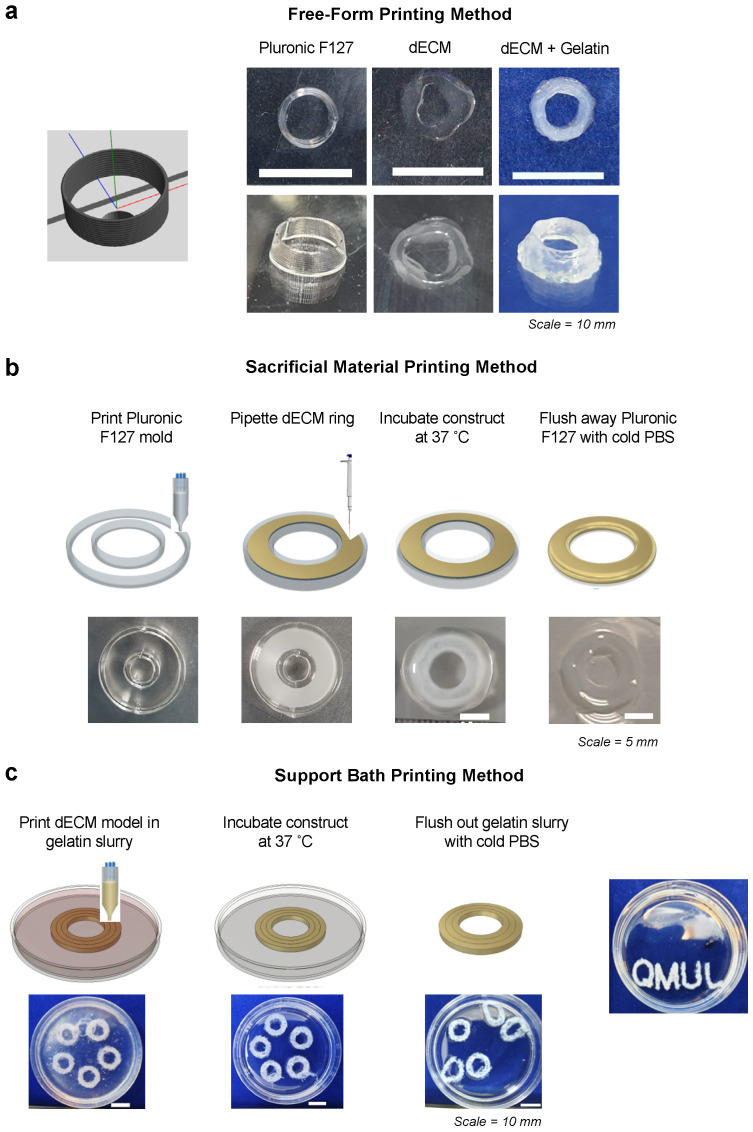
Compatibility of dECM with advanced biofabrication methods. (**a**) dECM was blended with 20% gelatin at a 1:1 ratio and was printed in a cylindrical shape at 29 °C. A total of 10 layers of a 3 mm diameter circle were printed, achieving a height of 2.5 mm. Scale bars represent 10 mm. (**b**) 25% Pluronic F127 was printed as the sacrificial ink in a ring shape template, as designed in BioCAD™. Ten layers of Pluronic were printed and approximately 150 μL of cold neutralized dECM was cast within the template. The model was incubated at 37 °C for 1 h to set the dECM and the Pluronic flushed away with cold 1X PBS. Scale bars represent 5 mm. (**c**) Concentric rings of dECM were printed into a 4.5% gelatin microparticle support bath. Ten layers of each ring were printed in clockwise and anti-clockwise directions and the dish was incubated at 37 °C for 1 h to set the dECM and melt the gelatin slurry. The gelatin was carefully pipetted away and flushed out with cold 1X PBS, leaving behind a dECM ring. Using this support bath method, we demonstrate a more complex design ‘QMUL’ can be achieved. Scale bars represent 10 mm.

**Figure 5 biomolecules-12-00837-f005:**
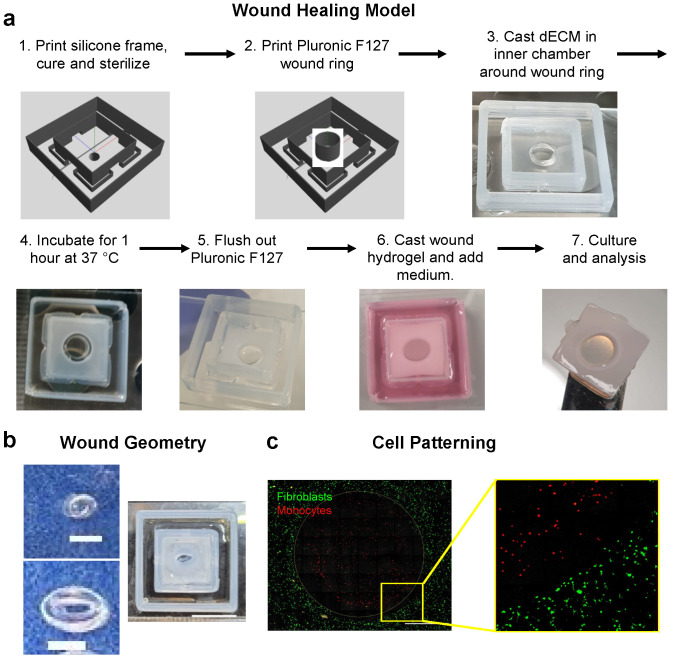
Application dECM in 3D printed wound healing model. (**a**) Schematic of fabrication of 3D printed wound healing model consisting of 3D printed silicone (SE1700) inner (12 mm × 12 mm) and outer (24 mm × 24 mm) frames with 2 mm wide x 4 mm tall gaps in the inner frame. A model wound is created by 3D printing a sacrificial Pluronic F127 ring (3 mm diameter), casting dECM hydrogels around the ring, and removing the ring with cold medium. A fibrin gel can then be cast in the wound and cultured in cell culture medium. (**b**) Example of different wound geometries consisting of ellipses (1.0 mm × 0.5 mm upper or 2.5 mm × 2.0 mm lower and right). Scale bars equal 1 mm left and 10 mm right. (**c**) Fluorescence images of monocytes and fibroblasts labeled with Cellbrite membrane dyes in the fibrin wound or dECM tissue compartments, respectively. Scale bar equals 1 mm.

## Data Availability

Not applicable.

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
