# Peer review of "Multi-Scale Analysis of the Composition, Structure, and Function of Decellularized Extracellular Matrix for Human Skin and Wound Healing Models"

_biomolecules, 2022, doi:10.3390/biom12060837_

Round 1
Reviewer 1 Report
I am fine with the publication of the manuscript
Author Response
We thank the reviewer for the positive assessment.
Reviewer 2 Report
Dear Authors,
please, improve description about room temperature, printing bed temp and hydrogel temp. during printing in the material and methods section 2.10.
Some additional reference:
https://doi.org/10.1088/1748-605X/abe55e
https://doi.org/10.1038/s41598-020-60196-y
After these minor revisions, in my opinion the paper could be acepted.
Author Response
We thank the reviewer for the positive assessment and recommendations for final edits. We have added printing temperature details in the materials and methods, and we have cites the suggested references in the main text.
This manuscript is a resubmission of an earlier submission. The following is a list of the peer review reports and author responses from that submission.
Round 1
Reviewer 1 Report
In their manuscript, Sarmin et al. formulated an extrudable ink of decellularized porcine skin ECM and characterized this new material via several biochemical and biological methods. The manuscript is an addition to the previously published skin dECM-derived skin equivalent models and succeeded to add some new mass spectrometry data to the existing knowledge. The authors claimed that the dECM has been incompletely understood, and this is why they now aimed to perform in-depth dECM characterization. Unfortunately, however, the characterization did not go much deeper than the high number of already published articles on dECM. Some more depth would greatly improve the novelty and significancy of this manuscript. Particularly unfortunate is that it remains unanswered, why the dECM hydrogels supported the growth of a thicker epidermal layer compared to the collagen/Matrigel hydrogels, considering that this was one of the main new results in the manuscript. In overall, the current manuscript remained too superficial to really increase the understanding of dECM inks. Clearly some additional characterization needs to be done in order to fulfill the promises made in the title and the abstract of the manuscript. The current manuscript does not meet the standards of Biomolecules and we believe that there would be journals better suited to this manuscript.
For future improvements of the manuscript, our more detailed comments are as follows:
- As the pepsin was not inactivated after the digestion step, was not there a risk of further digestion during and after the gelation of the material?
- What was the origin and activity of pepsin?
- In Figure 2e, the DNA content of a native skin could be useful.
- The authors had problems with the low viscosity of their ink. It really was exceptionally low, and this should be discussed with comparison to the previous literature. Furthermore, it was not clear, why the authors did not increase the viscosity just by increasing the polymer concentration. Also, the digestion time in the pepsin solution (3-5 days) was longer than typical in previously published studies (1-2 days). As the long digestion time is known to decrease the viscosity, it would be beneficial to study the effect of the digestion time on the viscosity.
- Based on previous literature and our experience, the stiffness of the dECM in the manuscript was extremely low (several pascals instead of several tens or hundreds of pascals) and made us wonder, how such a very soft gel remains intact during handling and cell seeding. The authors should discuss the mechanical integrity of their hydrogels and compare the stiffness values to the previously published values.
- In Figure 5b, the epidermal thickness of a native skin would be needed.
- Was the 3D printing done at room temperature? Besides increasing the concentration, also increasing the temperature might have helped the authors to achieve a more suitable viscosity. The viscosity should be measured at different temperatures to find out the best printing temperature.
- In Figure 6, the color of the dECM hydrogels seemed to vary from fully transparent to white. As also the dECM/gelatin hydrogels seem to be white, it may be that the dECM gels printed in gelatin bath are white because of the residual gelatin trapped in the dECM gels. This should be studied and discussed in the manuscript.
- Authors claimed that their skin dECM is compatible with a range of biofabrication methods. However, in the free-form printing the shape fidelity of the dECM ink without gelatin was really poor. This unfortunately did not support the conclusion of a broad compatibility.
Reviewer 2 Report
Dear Author, too many parts of the manuscript are already published in the following article, https://currentprotocols.onlinelibrary.wiley.com/doi/full/10.1002/cpz1.393 In my opinion, actually, your work should be converted in a Communication only around 3D technologies, adding in-vitro cells tests for a functional application (tissue regeneration: endothelial cells, umbilical cells,...) Only after these strong addictions, in my opinion, will it be possible to resubmit the document for the consideration this this prestigious magazine (IF 4.57)